# Association between Locomotive Syndrome and Hearing Loss in Community-Dwelling Adults

**DOI:** 10.3390/jcm12175626

**Published:** 2023-08-29

**Authors:** Sadayuki Ito, Hiroaki Nakashima, Naoki Segi, Jun Ouchida, Shinya Ishizuka, Yasuhiko Takegami, Tadao Yoshida, Yukiharu Hasegawa, Shiro Imagama

**Affiliations:** 1Department of Orthopedic Surgery, Nagoya University Graduate School of Medicine, Nagoya 466-8560, Aichi, Japan; sadaito@med.nagoya-u.ac.jp (S.I.); naoki.s.n@gmail.com (N.S.); orthochida@gmail.com (J.O.); shinyai@med.nagoya-u.ac.jp (S.I.); takegami@med.nagoya-u.ac.jp (Y.T.); imagama@med.nagoya-u.ac.jp (S.I.); 2Department of Otorhinolaryngology, Nagoya University Graduate School of Medicine, Nagoya 466-8560, Aichi, Japan; tadaoy@med.nagoya-u.ac.jp; 3Department of Rehabilitation, Kansai University of Welfare Science, Kashiwara 582-0026, Osaka, Japan; hasegawa@tamateyama.ac.jp

**Keywords:** hearing loss, locomotive syndrome, motor function, gait speed

## Abstract

The relationship between hearing and motor function as a function of aging is unclear. Therefore, we aimed to clarify the relationship between age-related hearing loss and locomotive syndrome. In total, 240 participants aged ≥40 years, whose hearing acuity and motor function had been measured, were included in this study. Patients with a hearing acuity of <35 dB and ≥35 dB were categorized into normal and low hearing acuity groups, respectively. Motor function was compared according to sex between the groups. Among men, those in the low hearing acuity group (51/100) were older, had a significantly slower walking speed, and had a higher prevalence of locomotive syndrome than those in the normal group. Among women, those in the low hearing group (14/140) were older and had a significantly slower gait speed than those in the normal group. The multivariate analysis showed that, in the low hearing acuity group, age and gait speed were risk factors in men, while age was the only risk factor in women. In conclusion, hearing loss was associated with walking speed. The association between hearing loss and locomotive syndrome was observed only in men. In the multivariate analysis, hearing loss was associated with walking speed only in men.

## 1. Introduction

Sensory organs have a significant role in motor function [1]. Therefore, when considering motor function decline, sensory organs should be evaluated in addition to muscle strength. Although motor function tends to be the focal point with regard to the risk of falls in older adults, sensory organs also play an important role [2]. Thus, it is of great importance to understand the relationship between motor function and sensory organs from the perspective of motor function decline and falls. Although there have been several reports on the relationship between motor function and sensory organs, this relationship remains unclear [1].

Hearing has been shown to have a significant effect on motor function [1]. Several studies have shown that older adults with hearing impairment have increased functional disability, decreased the ability to perform activities of daily living (ADL), and an increased risk of frailty [3,4]. Motor and auditory function decline with age [5,6], and both motor and auditory function decline have been reported to be associated with atherosclerosis [7,8]. Thus, common factors that affect motor and auditory functions have been observed and are closely related.

Locomotive syndrome (LS) is an important indicator of motor function [9]. LS significantly reduces the quality of life (QOL) [10] and shortens the life expectancy. Moreover, the prevention of LS has long been advocated to maintain and improve physical function in middle-aged and older adults [11].

To our knowledge, no study has reported an association between auditory function and LS, an important assessment of motor function. Once these associations are clarified, an association between hearing and locomotor function can be established, and effective prevention and treatment methods for motor function decline can be considered. This could significantly reduce the number of people requiring assistance and care in their daily lives.

The prevalence of hearing loss increases with age. Hearing loss begins in the 40s and is present in 50–80% of people in their 80s [6]. It has been reported that hearing loss is more common in men [6], and LS is more common in women [12]. Therefore, a comparison of hearing and motor function, including LS, in men and women over the age of 40 years would help to clarify the relationship between hearing and motor function as a function of aging. Therefore, this study aimed to evaluate this relationship according to age and sex in participants aged ≥40 years using a large cohort and comparing the differences between the two groups.

## 2. Materials and Methods

### 2.1. Study Participants

Participants in the study voluntarily took part during municipally sponsored health screenings in the town of Yakumo in 2019. The town of Yakumo boasts a population of around 17,000 residents, with nearly 28% being 65 years or above. Unlike city regions, Yakumo’s population is prominently involved in farming and fishing activities. Health screenings have been a consistent event in Yakumo since 1982, with the health checks comprising optional bone health checks, functional physical evaluations, internal health checks, mental evaluations, and a general health quality assessment (SF-36) [13]. For this particular research, we considered participants who underwent evaluations for auditory capacity, bioelectrical impedance analysis (BIA), spinal strength, walking pace, and balance tests. Individuals suffering from significant knee traumas, advanced hip osteoarthritis, nervous system ailments, profound mental health issues, and serious walking or standing impairments were not considered for the study.

Of the 537 participants who underwent physical examination, 240 (100 men and 140 women) met the inclusion criteria (Figure 1). The study protocol was approved by the University’s Human Research Ethics Committee and Institutional Review Board. All study participants gave written informed consent prior to participation. The study was conducted in accordance with the principles of the Declaration of Helsinki.

### 2.2. Hearing Acuity Test

An otolaryngologist examined the ears, nose, and throat before conducting the hearing tests. The hearing levels were evaluated using an audiometer (Model AA-79S; Rion, Tokyo, Japan) in a quiet room. We measured the hearing levels bilaterally at 1 and 4 kHz. The World Health Organization defines “disabling” hearing loss as hearing loss greater than 35 decibels (dB) in the better-hearing ear [14]. Participants with a hearing acuity of 34 dB and below at 4 kHz in the better-hearing ear were categorized into the normal hearing group (N group), and those with a hearing acuity of 35 dB and above were categorized into the hearing loss group (L group).

### 2.3. Measurement through Bioelectrical Impedance

Physical dimensions such as height, weight, BMI, and the muscle mass index for each extremity were gauged with the BIA method. The InBody 770 BIA machine from Inbody Co., Ltd., Seoul, the Republic of Korea, which identifies tissues by their electric resistance, was employed [15]. The formula for BMI is given as: BMI = weight in kg/square of height in meters. The BIA process directly estimates the muscular mass for each extremity, and the formula for SMI is SMI = limb skeletal muscle mass in kg/square of height in meters.

### 2.4. Motor Function Evaluation

#### 2.4.1. Strength of Back Muscles

The utmost isometric force from the back’s extensor muscles represents back muscle power. This was gauged using a T.K.K. 5102 digital dynamometer from Takei Seisakusho, Tokyo, Japan, while the subject stood bending their lumbar at 30° and extending their knee. The average of two separate tests was noted, and the highest force for each test was taken into account [16].

#### 2.4.2. Speed of Walking

The speed of mobility was deduced from the time a participant needed to traverse a 10 m straight path at their quickest speed. Each participant was evaluated twice, and the average of these times was utilized for evaluation [17].

#### 2.4.3. Balance Analysis

The Gravicorder GW-5000 by Anima Co. Ltd., Japan, was employed to gather data on balance. Participants were instructed to stand at the apparatus’s midpoint with both feet touching for one minute, first with eyes open and then eyes shut. The tool uses force sensors to detect any immediate shifts in the pressure’s center point (COP). The evaluated outcomes comprised the full distance of COP movement and the average squared area covered by COP shifts. Data were registered at a frequency of 100 Hz [18]. The metrics utilized included COP displacement per second and the encompassing area of the COP shift.

### 2.5. Locomotive Syndrome

As per the guidelines of the Japanese Orthopaedic Association (JOA), LS risk is gauged with three specific tests: a dual-step exam, a standing from seated test, and the GLFS-25 questionnaire [17,19]. These evaluations categorize LS into two stages. The first stage suggests the initiation of declining motor function, whereas the second stage points towards an ongoing deterioration of motor skills.

The JOA-prescribed evaluations were executed as follows [17,19]. The standing test measured the capability to rise from a seated position on varying height platforms: 40 cm, 30 cm, 20 cm, and 10 cm in sequence. Each level was ranked based on difficulty, with the outcomes based on the minimum stool height a participant could stand from without assistance. In the dual-step test, the therapist measured the span of two strides starting from a fixed point to the toes’ tips. The results were determined by adjusting the largest two-step length against the participant’s height.

GLFS-25 is an exhaustive self-assessment referring to the past 30 days. This questionnaire incorporates four pain-related questions, sixteen daily activity-related queries, three questions about societal interactions, and two mental health-related questions. The grading for each response ranges from no issues (0 points) to significant issues (4 points).

Height, weight, body mass index (BMI), and skeletal muscle mass index (SMI) measured by BIA (InBody 770 BIA device, Inbody Co., Seoul, Korea) were used [15]. The BMI and SMI were calculated as follows: BMI = weight (kg)/height^2^ (m^2^), SMI = appendicular skeletal muscle mass (kg)/height^2^ (m^2^).

The LS stages were rated on a scale of 0, 1, and 2 [17,19]. In this study, LS0 was defined as normal (N) and LS1 and 2 as LS patients. LS0 was defined as being able to stand on one leg (both legs) from a seat 40 cm high, with a 2-step test score of 1.3 or higher, and a 25-question GLFS score of less than 7 points, in which case the patient was defined as N. If this was not met, the case was designated as LS.

### 2.6. Statistical Analysis

Continuous variables were compared between groups L and N by the Student’s *t*-test, and categorical variables were compared between groups L and N by the chi-square test. Analyses were performed separately for men and women. Logistic regression analysis was used to verify the presence of risk factors in the L group. Univariate analysis was performed with variables that showed a *p*-value < 0.05. Using SPSS Statistics software (version 29.0; IBM Corp.), a *p*-value < 0.05 was considered statistically significant.

## 3. Results

The total number of participants was 240 (100 males and 140 females, mean age 63.9 ± 9.6 years); 175 were in the N group and 65 in the L group (Table 1).

### 3.1. Male Participants

The mean age of the male participants was 66.0 ± 8.8 years, 49 in group N and 51 in group L. The mean SMI was 7.81 ± 0.66 kg/m^2^. The mean back muscle strength was 113.1 ± 24.4 kg. The mean walking speed was 2.18 ± 0.38 m/s (Table 1).

Univariate analysis revealed significant differences in age, gait speed, and the ratio of Locomotive Syndrome (N/LS) (age; N: 63.0 ± 8.7, L: 68.9 ± 7.9, *p* < 0.001, gait speed; N: 2.30 ± 0.34, L: 2.06 ± 0.39, *p* = 0.002, LS ratio (N/LS); N: N/LS = 32/17, L: N/LS = 19/32; *p* = 0.006). No significant differences were found in other items (Table 2).

Logistic regression analysis of the age, walking speed, and LS as covariates for risk factors in group L revealed that age and walking speed were risk factors in group L (age: Exp (B) 1.076, 95% confidence interval (CI): 1.017–1.140, *p* = 0.011; gate speed: Exp (B) 0.254, 95% CI: 0.067–0.96, *p* = 0.043) (Table 3).

### 3.2. Female Participants

The mean age of the female participants was 62.4 ± 9.8 years, 126 in group N and 14 in group L. The mean SMI was 6.23 ± 0.73 kg/m^2^. The mean back muscle strength was 65.1 ± 58.6 kg. The mean walking speed was 1.99 ± 0.32 m/s (Table 1).

Univariate analysis revealed significant differences in age and gait speed (N: 61.3 ± 9.6, L: 72.0 ± 6.4, *p* < 0.001, gait speed; N: 2.01 ± 0.32, L: 1.83 ± 0.23, *p* = 0.036). No significant differences were found for the other items (Table 4).

Logistic regression analysis of age and gate speed as covariates for risk factors in group L showed that age was the only risk factor in group L (Exp (B) 1.176, 95% CI: 1.066–1.296, *p* = 0.001) (Table 5).

## 4. Discussion

Several studies on motor function and hearing have reported that older adults with hearing impairment have increased functional disability, a decreased ability to perform ADL, and an increased risk of frailty [3,4]. However, to our knowledge, no studies have reported an association between hearing and LS. The present study is the first to examine the relationship between LS and hearing loss. We found an association between hearing loss and walking speed. In men, the prevalence of LS was higher in participants with hearing loss. However, when adjusted for age, only gait speed was associated with hearing loss in men.

Older adults with hearing loss tend to exhibit mobility impairments such as slow walking [20]. In the present study, the walking speed was reduced in participants with hearing loss, which is consistent with previous reports. A possible explanation for the association between hearing loss and slower gait is poor balance related to hearing loss [20]. This may be due to the limited access to static auditory cues, which provide a spatial reference for comparing locations. The anchoring effect of auditory cues is important for the maintenance of motor and postural controls [21]. When cues are not available owing to reduced auditory acuity, the body balance and gait control are disrupted, resulting in increased gait variability. In addition, cochlear and vestibular dysfunction (i.e., inner ear problems) may coexist in people with hearing loss [22], and increased gait variability may be observed as a result of vestibular dysfunction [23].

In this study, an association was found between LS and hearing loss only in men. LS is defined as “a condition in which motor function is impaired due to motor impairment” and often requires long-term care services for the elderly [17]. LS leads to a high degree of disability that interferes with daily life and reduces the QOL [10]. In LS, the decline in motor skills makes mobility difficult, and the ability to recover from stress is reduced. Recent studies have argued that LS extends the concept of physical frailty and that preventing LS can prevent physical frailty and subsequent disability [24]. Therefore, although various factors affecting LS have been investigated, this is the first study to show an association between LS and hearing loss in women. The prevalence of LS is higher in women than in men [16]. This may be due to the higher prevalence of musculoskeletal disorders in women than in men and lower muscle strength and motor function. It is possible that the prevalence of LS in men is influenced by factors other than locomotor disease, as in women, and that hearing loss, as revealed in the present study, may be a contributing factor. It is also possible that vestibular function loss associated with hearing loss affects LS in men and leads to a decreased walking speed.

The rate of hearing loss was higher in men [6]. The reason for this is that men are more affected by occupational and impulse noise exposure than women [25]. Other reasons have been reported regarding the effects of sex hormones, with estrogen having a positive effect on auditory information processing and testosterone having a negative effect on auditory performance [26]. The difference in the prevalence of hearing loss decreases with age, as androgens decline with age in men and estrogen declines with age in women, suggesting that hearing impairment is more common in older women than in men. Regarding motor function, women are inherently not as strong as men, and they have a higher prevalence of LS [12]. In addition, men have a higher rate of age-related loss of muscle mass, and the muscle mass difference between men and women decreases with age [27]. In the present study, men had a higher rate of hearing loss, and women had weaker muscles and a higher prevalence of LS. An association was observed between hearing loss and gait speed in both men and women, and the association between hearing loss and LS was observed only in men. Since hearing loss was observed earlier in men than in women, and motor function decline was observed at older ages, the results of the present study, which included individuals in the early stages of LS, suggest that women tend to develop LS before hearing loss, whereas men may not develop LS until the occurrence of hearing loss in addition to muscle weakness.

It has been previously reported that individuals with cardiovascular risk factors such as hypertension, diabetes, smoking, increased serum cholesterol, or a decreased estimated glomerular filtration rate are at risk of developing hearing impairment [28]. Previous studies have reported that hypertension and smoking are risk factors for LS [17]. In addition, both LS and hearing loss are reportedly associated with oxidative stress [29]. We believe that lifestyle modifications, such as dietary modifications, including salt restriction and antioxidant intake, increased physical activity, and smoking cessation, may reduce the risk of LS and hearing loss. A previous study reported that regular exercise significantly delays age-related hearing loss and degeneration of the cochlea [30]. A set of exercises for LS called locomotion training (LT), which are mainly aimed at strengthening the muscles of the lower limbs and improving balance skills essential for walking and basic ADL, has been proposed to prevent LS. Each exercise has been proven effective, and we believe that regular LT is effective in preventing hearing and motor function loss [31].

This study has several limitations. First, the participants were middle-aged and older adults living in relatively rural areas, many of whom engaged in farming and fishing. Thus, their lifestyle differs from those of adults in urban areas. Therefore, the results of this study reflect only results in a specific demographic population and its relevance to the broader population cannot be confirmed. Therefore, the results do not reliably suggest that a direct relationship exists between hearing loss and motor function. Furthermore, although the preliminary results showing an association between hearing loss and motor function were revealed by multivariate analysis, this is part of hypothesis generation, and we cannot ignore the possibility that these associations may have occurred by chance. In other words, our observations could be a type 1 error. Therefore, our results require additional, more rigorous studies to confirm the association between these variables. Second, they attended annual health checkups, suggesting that they may be more health-conscious than others. In addition, less than half (240) of the total 537 participants in 2019 received all of the required screening items this time, suggesting that only health-conscious people may have been considered. Third, the rate of hearing loss was 14 out of 140 women, which is a small number of cases, and should be examined and confirmed using a larger population in the future. Finally, this was a cross-sectional single-site study. Future longitudinal, multicenter studies are required to validate our findings.

## 5. Conclusions

Hearing loss and motor function were compared in healthy participants using the community health examination data. Hearing loss was associated with walking speed. The association between hearing loss and LS was observed only in men. Additionally, in the multivariate analysis, hearing loss was associated with walking speed only in men.

## Figures and Tables

**Figure 1 jcm-12-05626-f001:**
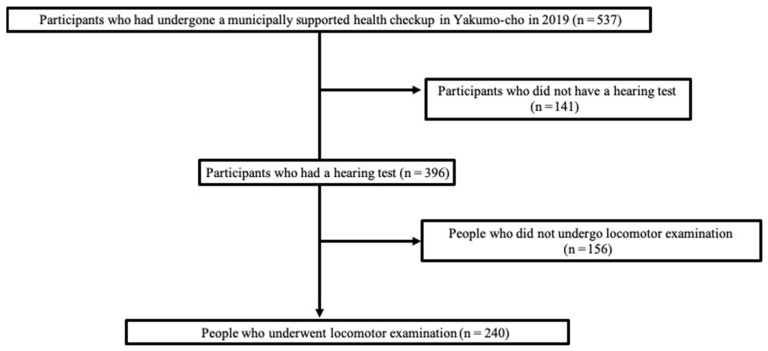
Flowchart of the patient selections.

**Table 1 jcm-12-05626-t001:** Comparison of each parameter between male and female participants.

	Total (*n* = 240)	Male (*n* = 100)	Female (*n* = 140)	*p*
Hearing acuity (N/L)	175/65	49/51	126/14	<0.001 *
Age (y)	63.9 ± 9.6	66.0 ± 8.8	62.4 ± 9.8	0.003 *
BMI (kg/m^2^)	23.8 ± 3.5	24.5 ± 2.9	23.2 ± 3.8	0.005 *
BFP (%)	29.1 ± 7.4	23.7 ± 4.2	33.0 ± 6.8	<0.001 *
SMI (kg/m^2^)	6.88 ± 1.05	7.81 ± 0.66	6.23 ± 0.73	<0.001 *
Back muscle strength (kg)	84.8 ± 53.1	113.1 ± 24.4	65.1 ± 58.6	<0.001 *
Gait speed (m/s)	2.07 ± 0.36	2.18 ± 0.38	1.99 ± 0.32	<0.001 *
Length of COP sway (eyes open) (cm/s)	1.63 ± 0.48	1.80 ± 0.49	1.50 ± 0.44	<0.001 *
Length of COP sway (eyes closed) (cm/s)	2.04 ± 0.85	2.43 ± 0.90	1.76 ± 0.68	<0.001 *
Surrounding area of COP sway (eyes open) (cm^2^)	2.64 ± 1.48	2.89 ± 1.45	2.46 ± 1.48	0.030 *
Surrounding area of COP sway (eyes closed) (cm^2^)	3.30 ± 2.47	4.01 ± 2.55	2.79 ± 2.29	<0.001 *
Locomotive Syndrome (N/LS)	103/137	51/49	52/88	0.035 *

Hearing acuity (N/L): normal hearing group (N group)/hearing loss group (L group), BMI: body mass index, BFP: body fat percentage, SMI: skeletal muscle mass index, COP: the center of pressure, Locomotive Syndrome (N/LS): participants with LS0 (N)/participants with LS1 and 2 (LS), *: *p* < 0.05.

**Table 2 jcm-12-05626-t002:** Comparison of each parameter between the N and L groups of male participants.

Male	N (*n* = 49)	L (*n* = 51)	*p*-Value
Age (y)	63.0 ± 8.7	68.9 ± 7.9	<0.001 *
BMI (kg/m^2^)	24.5 ± 2.9	24.6 ± 2.9	0.878
BFP (%)	23.6 ± 4.1	23.8 ± 4.3	0.794
SMI (kg/m^2^)	7.84 ± 0.6	7.78 ± 0.71	0.68
Back muscle strength (kg)	113.2 ± 24.3	113 ± 24.8	0.968
Gait speed (m/s)	2.30 ± 0.34	2.06 ± 0.39	0.002 *
Length of COP sway (eyes open) (cm/s)	1.72 ± 0.44	1.89 ± 0.52	0.087
Length of COP sway (eyes closed) (cm/s)	2.29 ± 0.79	2.56 ± 0.99	0.141
Surrounding area of COP sway (eyes open) (cm^2^)	2.72 ± 1.11	3.05 ± 1.71	0.255
Surrounding area of COP sway (eyes closed) (cm^2^)	3.6 ± 2.16	4.4 ± 2.84	0.123
Locomotive Syndrome (N/LS)	32/17	19/32	0.006 *

Hearing acuity (N/L): normal hearing group (N group)/hearing loss group (L group), BMI: body mass index, BFP: body fat percentage, SMI: skeletal muscle mass index, COP: the center of pressure, Locomotive Syndrome (N/LS): participants with LS0 (N)/participants with LS1 and 2 (LS), *: *p* < 0.05.

**Table 3 jcm-12-05626-t003:** Logistic regression analysis for risk factors of hearing loss (L group) in male participants.

Male	B	SE	Wald	df	*p*	Exp (B)	95% CI
Age	0.074	0.029	6.398	1	0.011 *	1.076	1.017–1.140
Gait speed	−1.371	0.678	4.082	1	0.043 *	0.254	0.067–0.96
Locomotive Syndrome (N/LS)	0.854	0.453	3.56	1	0.059	2.349	0.967–5.705

Locomotive Syndrome (N/LS): participants with LS0 (N)/participants with LS1 and 2 (LS), *: *p* < 0.05.

**Table 4 jcm-12-05626-t004:** Comparison of each parameter between the N and L groups of the female participants.

Female	N (*n* = 126)	L (*n* = 14)	*p*-Value
Age (y)	61.3 ± 9.6	72.0 ± 6.4	<0.001 *
BMI (kg/m^2^)	23.1 ± 3.9	24.7 ± 3.3	0.137
BFP (%)	32.7 ± 6.9	35.8 ± 5.2	0.119
SMI (kg/m^2^)	6.23 ± 0.73	6.2 ± 0.71	0.872
Back muscle strength (kg)	66 ± 61.6	57.3 ± 11.5	0.602
Gait speed (m/s)	2.01 ± 0.32	1.83 ± 0.23	0.036 *
Length of COP sway (eyes open) (cm/s)	1.5 ± 0.46	1.51 ± 0.19	0.956
Length of COP sway (eyes closed) (cm/s)	1.75 ± 0.7	1.86 ± 0.4	0.567
Surrounding area of COP sway (eyes open) (cm^2^)	2.5 ± 1.53	2.16 ± 0.92	0.428
Surrounding area of COP sway (eyes closed) (cm^2^)	2.82 ± 2.39	2.52 ± 1.2	0.654
Locomotive Syndrome (N/LS)	48/78	4/10	0.571

Hearing acuity (N/L): normal hearing group (N group)/hearing loss group (L group), BMI: body mass index, BFP: body fat percentage, SMI: skeletal muscle mass index, COP: the center of pressure, Locomotive Syndrome (N/LS): participants with LS0 (N)/participants with LS1 and 2 (LS), *: *p* < 0.05.

**Table 5 jcm-12-05626-t005:** Logistic regression analysis for risk factors of hearing loss (L group) in the female participants.

Female	B	SE	Wald	df	*p*	Exp (B)	95% CI
Age	0.162	0.05	10.549	1	0.001 *	1.176	1.066–1.296
Gait speed	−0.924	1.001	0.851	1	0.356	0.397	0.056–2.825

*: *p* < 0.05.

## Data Availability

The health checkup data used to support the findings of this study are available from the corresponding author upon request.

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
