# Peer review of "Association between Locomotive Syndrome and Hearing Loss in Community-Dwelling Adults"

_jcm, 2023, doi:10.3390/jcm12175626_

Round 1
Reviewer 1 Report
The aim of this study was to evaluate the relationship between hearing loss and LS according to age and sex in the survey participants aged ≥40 years. I appreciate the research area that the authors have chosen to focus on. I have reviewed this well-written paper with great interest. I have some recommendations for this paper:
(1) Many participants were excluded, and thus I would like to know their influence and generalization of the results. Please show a figure of the patient recruitment and discuss the selection bias in limitation section.
(2) I would like to know the differences of the parameters between men and women, as the authors conducted analysis separately for men and women. Please add the analyses in Table 1.
(3) Page 6, Line 214-218, “Several studies on motor function and hearing have reported that older adults with hearing impairment have increased functional disability, decreased ability to perform ADL, and an increased risk of frailty [3,4]. However, the present study is the first to report an association between motor function and hearing, including in LS.”
If there are similar reports in the past, I think it is not "first". Please explain this discrepancy in detail.
(4) There are only 14 women in the L group, which may lack statistical power. Could this be one of the reasons why there was no association in women? Please discuss statistical power in discussion section.
Reviewer 2 Report
This is a cross-sectional, case-control study that compared patients older than 65 years with and without hearing impairment in relation to various aspects of motor skills, especially gait and balance. The main hypothesis was that hearing loss could be associated with changes in motor skills. The main objective was to verify whether the presence of locomotor syndrome would be associated with hearing loss. The authors evaluated 240 subjects and the main finding was that those with low hearing were older and had lower gait speed. It was not possible to confirm the relationship between hearing loss and the presence of locomotor syndrome. The study design has many limitations in itself, for the purposes of the research, and the changes observed are the same as those expected from the effect of age: greater hearing loss and slower gait. Unfortunately, there are no new or relevant observations in this study.
None
